# Aesthetic evaluation and the perceived properties of Chinese characters

**Qinjing Zhang** [1] *, **Hiroyuki Mitsudo** [2]

**1** Graduate School of Human-Environment Studies, Kyushu University, Fukuoka, Japan, **2** Faculty of Human-Environment Studies, Kyushu University, Fukuoka, Japan

* zhang.qinjing.156@s.kyushu-u.ac.jp

**Data Availability Statement:** All data files are available from the OSF database (URL: https://osf.io/mbqzg/).

**Funding:** This work was supported by JST SPRING, Grant Number JPMJSP2136 and Research Grants for Degree Completion (Doctoral

## Abstract

Previous studies have indicated that the visually perceived properties of geometrical figures influence aesthetic evaluations. However, it is unclear whether findings based on geometrical figures can be extended to artworks such as paintings and calligraphy, as artworks have their own contents and meanings. To answer this question, we designed experiments in which users of Chinese characters were asked to rate the perceived symmetry, complexity, prototypicality, and beauty of Chinese characters. Each character was presented to 35 Japanese and Chinese participants in five typing fonts in a laboratory setting (Experiment 1), and to 100 Japanese participants in five calligraphic handwriting scripts online (Experiment 2). By analyzing the relative impacts of perceived properties on aesthetic evaluation and their interactions with script styles using a generalized linear mixed model, we discovered that script style plays an important role in the association between the perceived properties and aesthetic evaluation of Chinese characters. These results are discussed in relation to studies on the aesthetic evaluation of geometrical figures and recent studies using Chinese calligraphy.

## Introduction

### Aesthetic evaluation and the perceived properties of visual patterns

The aesthetic evaluation of artwork and products is a common activity in daily life and can change our preferences and choices. The perceived properties of objects are determinants of aesthetic evaluation. Experimental aesthetic studies using abstract geometrical patterns have revealed that symmetry and complexity are important for aesthetic evaluation. Visual symmetry can be defined in various forms, including mirror and rotational symmetry. Regardless of the type of symmetry, experimental studies suggest that aesthetic evaluation improves when a stimulus is perceived as symmetrical [1]. In contrast to the case of symmetry, the relation between complexity and aesthetic evaluation is less straightforward [2]. Some studies suggest an inverted U-shape relationship between complexity and aesthetic evaluation, where aesthetic evaluation increases when the stimulus complexity is intermediate [3]. However, other studies have suggested that aesthetic evaluation increases when a stimulus is perceived as complex [1,4].

Program) of Kyushu University and QZ received these two awards. JST SPRING https://k-spring.kyushu-u.ac.jp/ Grants for Degree Completion (Doctoral Program) of Kyushu University https://www.hues.kyushu-u.ac.jp/news/%E4%BB%A4%E5%92%8C%EF%BC%95%E5%B9%B4%E5%BA%A6-%E3%80%8E%E5%AD%A6%E4%BD%8D%E5%8F%96%E5%BE%97%E8%AA%B2%E7%A8%8B%E5%8D%9A%E5%A3%AB%E3%81%AB%E5%90%91%E3%81%91%E3%81%A6%E3%81%AE%E7%A0%94%E7%A9%B6/ Sponsors did not play any role in this study.

**Competing interests:** The authors have declared that no competing interests exist.

Prototypicality, being a measure of how representative an object is of a category, is influenced partly by familiarization and has also been shown to influence aesthetic evaluations [5]. According to Martindale and Moore, stimuli with high prototypicality are preferred to those with lower prototypicality in aesthetic evaluations [6]. In line with this idea, Zajonc found that the repeated presentation of unfamiliar characters and faces (expected to produce familiarization) increases preference ratings [7].

## Visual object category

Visual object category, including geometrical figures, paintings, faces, and characters, plays an important role in aesthetic evaluation for some reasons [8]. First, object category modifies the way in which experience alters aesthetic evaluation. For example, familiarization with faces increases aesthetic evaluations, whereas familiarization with natural scenes decreases aesthetic evaluations [9,10]. Similar results have been reported for the relationship between balance and aesthetic judgements for single- and multi-element abstract patterns [11]. Second, in contrast to geometrical figures, artworks have their own meaning, which can evoke a strong aesthetic experience. On one hand, geometrical figures are commonly used to investigate cognitive processes for aesthetic evaluation because they can be easily manipulated in terms of the perceived properties of stimuli. However, from the perspective of aesthetic psychology, as advocated by Jacobsen, geometrical figures lack content [12] and are thus not optimal to evoke a strong aesthetic experience. On the other hand, artworks such as impressive paintings cannot be easily manipulated for a controlled psychophysical experiment, but have unique meanings and can evoke a strong aesthetic experience. Therefore, to understand the relationship between the perceived properties and aesthetic evaluation of artworks, it is useful to select materials whose perceived properties and meaning are easily manipulated.

## Chinese characters

This study focuses on a unique class of Chinese characters that convey meaning through their shapes. Chinese characters are hieroglyphs that combine symbolic visual representations of objects and things. Elementary parts of Chinese characters have ideographic origins. Whereas phonetic scripts such as the alphabet only contain sound features, ideographic characters are useful to convey meanings through visual image features. Therefore, Chinese characters can be regarded as stimuli that can not only be easily manipulated as patterns, but also carry meanings similar to visual artworks.

Chinese characters have been used as stimuli to investigate the mere exposure effect [7] and vary in complexity, as in the case of geometrical patterns [13]. In addition to the typing font of Chinese characters, previous studies have investigated the handwritten calligraphic scripts of Chinese characters. Calligraphy is a type of visual art that has been developed over thousands of years and conveys the aesthetics of characters and letters through handwriting. The influence of symmetry, balance, and prototypicality on aesthetic evaluation has been investigated using Chinese calligraphy artwork [14,15].

Previous studies have employed non-Chinese-character users, who treat Chinese characters as stimuli without attached meanings. Regarding the appreciation of Chinese calligraphy, meanings are of the same importance as the shape of the handwriting. In addition, the fonts used for typing Chinese characters may differ considerably from the styles employed in calligraphic scripts. Fillinger and Hübner examined prototypicality by asking non-Chinese-character users to determine whether calligraphic stimuli resembled Chinese characters [15]. However, the prototypicality of Chinese characters is not only linked to whether they resemble a Chinese character but also to whether the specific character in calligraphy is similar to the

typing font. Therefore, recruiting Chinese character users is necessary, especially in studies that use Chinese characters to investigate the mechanisms of aesthetic evaluation.

Two recent studies investigated the cognitive mechanisms underlying the aesthetic evaluation of meaningful Chinese characters. Using electroencephalography, Li et al. demonstrated that native Chinese speakers show different event-related potentials for liked and disliked Chinese characters [16]. Additionally, they revealed brain networks for understanding Chinese calligraphy [17]. However, the character stimuli used by Li et al. were controlled for complexity. Almost all calligraphy artworks used by Li et al. were judged to be relatively unfamiliar to the participants [17]. Therefore, the relationship between the perceived properties and aesthetic evaluation of Chinese characters remains unclear.

## Aims of the present study

The primary goal of this study was to elucidate whether the perceived properties claimed to influence the aesthetic evaluation of geometrical figures can predict the aesthetic evaluation of stimuli with meanings, such as Chinese characters. To achieve this goal, we designed two experiments in which Chinese character users were asked to rate the perceived properties and beauty of 18 Chinese characters with relatively neutral emotional valences. In an attempt to make this study comprehensive, the 18 characters were presented in five typing fonts (Experiment 1) and five calligraphic handwriting scripts (Experiment 2). We did not aim at examining the effect of meaning itself on perceived properties or aesthetic evaluation. This study focused on three perceived properties: symmetry, complexity, and prototypicality, each of which was rated by participants in Experiments 1 and 2. Because all the perceived properties mentioned above were claimed to have positive effects on the aesthetic evaluation of geometrical patterns in general, we hypothesized that they would have similar effects on Chinese character stimuli. In addition, we are interested in whether the results obtained using non-artistic fonts could be reproduced for artistic works in the context of Chinese characters.

## Experiment 1

### Methods

**Participants.**    Owing to the lack of previous research directly relevant to our purpose, we followed Brysbaert's advice to determine the sample size (d = 0.4, α = .05, and 1−β = .8) [18]. Using pwr package 1.3–0 running on R 4.0.5 [19,20], the power analysis indicated that at least 32 participants were required to obtain a power of .8. Consequently, we recruited 35 participants (21 females, 22.1 years old, SD = 4.10) from July 5 to 20, 2022 at Kyushu University and paid them 1,000 yen for their participation. All the participants were Chinese character users (17 native Japanese speakers and 18 native Chinese speakers). This study was approved by the Ethics Committee of the Faculty of Human-Environment Studies of Kyushu University (number: 2021–033). Written informed consent was obtained from all the participants.

**Apparatus and materials.**    The experiment was conducted in a sound-attenuating chamber. Stimuli were presented on a 21.9-inch (46 × 29 cm) LCD monitor (EIZO CG223W). The stimulus presentation and data collection were controlled using a personal computer (EPSON Endeavor MR4300E). Stimuli were displayed on the Chrome browser software using jsPsych [21] and were presented in black (luminance: 0.12 cd/m$^2$) at the center of the screen with a height of 200 pixels. The instructional text was black (0.12 cd/m$^2$), the choice frame was dark gray (9.10 cd/m$^2$), and the background was light gray (42.6 cd/m$^2$).

A set of 18 Chinese characters was selected based on the results of the pilot experiment (Fig 1). The selection criteria were that (a) the emotional valences of the characters were not greatly biased positive or negative, and (b) their overall aesthetic evaluations were similar between

| run | turf | extend | plastic | display | rate |
|---|---|---|---|---|---|
| 走 | 芝 | 延 | 塑 | 陳 | 率 |
| 2.3 | 2.3 | 1.8 | 2.0 | 1.8 | 2.2 |

| know | to | alone | field | dish | three |
|---|---|---|---|---|---|
| 存 | 宛 | 孤 | 場 | 皿 | 三 |
| 2.3 | 2.0 | 1.8 | 2.3 | 1.8 | 2.0 |

| twist | stick | marrow | central | present | fence |
|---|---|---|---|---|---|
| 捻 | 粘 | 髓 | 枢 | 呈 | 柵 |
| 1.7 | 1.8 | 2.2 | 2.2 | 2.3 | 2.0 |

**Fig 1. Chinese character stimuli used in Experiments 1 and 2.** Note: The English translation is shown above each character. The number below each character is the mean emotional valence rating obtained in the pilot experiment (1 = negative; 2 = neutral; 3 = positive).

Japanese and Chinese participants (i.e., the two groups comprised Chinese character users, but differed to some extent in their usage). No characters explicitly meant symmetry, complexity, prototypicality, or beauty. The characters were shown using five typing fonts: Gothic, Maru Gothic, Mincho, Regular, and Semi-cursive (Fig 2) because the five fonts are equally and widely used for Chinese character users in China and Japan, but might differ in their prototypicality. The Gothic, Maru Gothic, and Mincho scripts mainly appear on the press and online media, and the Regular and Semi-cursive scripts appear on textbooks and personal letters. The Maru Gothic script has rounder corners than the Gothic scripts; the Mincho script is similar to the roman script in western countries; the Semi-cursive script tends to contain more continuous strokes than the Regular script.

**Procedure.** The participants made two types of judgements in separate blocks: aesthetic and perceived property ratings. In the perceived property rating blocks, three tasks regarding to symmetry, complexity, and prototypicality were conducted separately. All the instructions were provided in Japanese.

The participants were first asked to rate the beauty of each stimulus using a 4-point Likert scale (1 = lowest, 4 = highest). After the aesthetic evaluation of all stimuli was completed, three perceived property tasks were performed on the same stimuli using a 4-point Likert scale: symmetry (1 = asymmetric, 4 = symmetric), complexity (1 = simple, 4 = complex), and prototypicality (1 = not typical, 4 = typical). The order of the three tasks was counterbalanced across participants. The experiment comprised 360 trials consisting of 18 Chinese characters × five font types × four task types. For each task, the participants performed two successive blocks. Five practice trials were conducted before each task. The presentation order of the stimuli was randomized across blocks and participants.

**Analysis.** To investigate the factors contributing to aesthetic evaluation, we used generalized linear mixed models (GLMM) running in R (version 4.0.5) with the lme4 package [20,22].

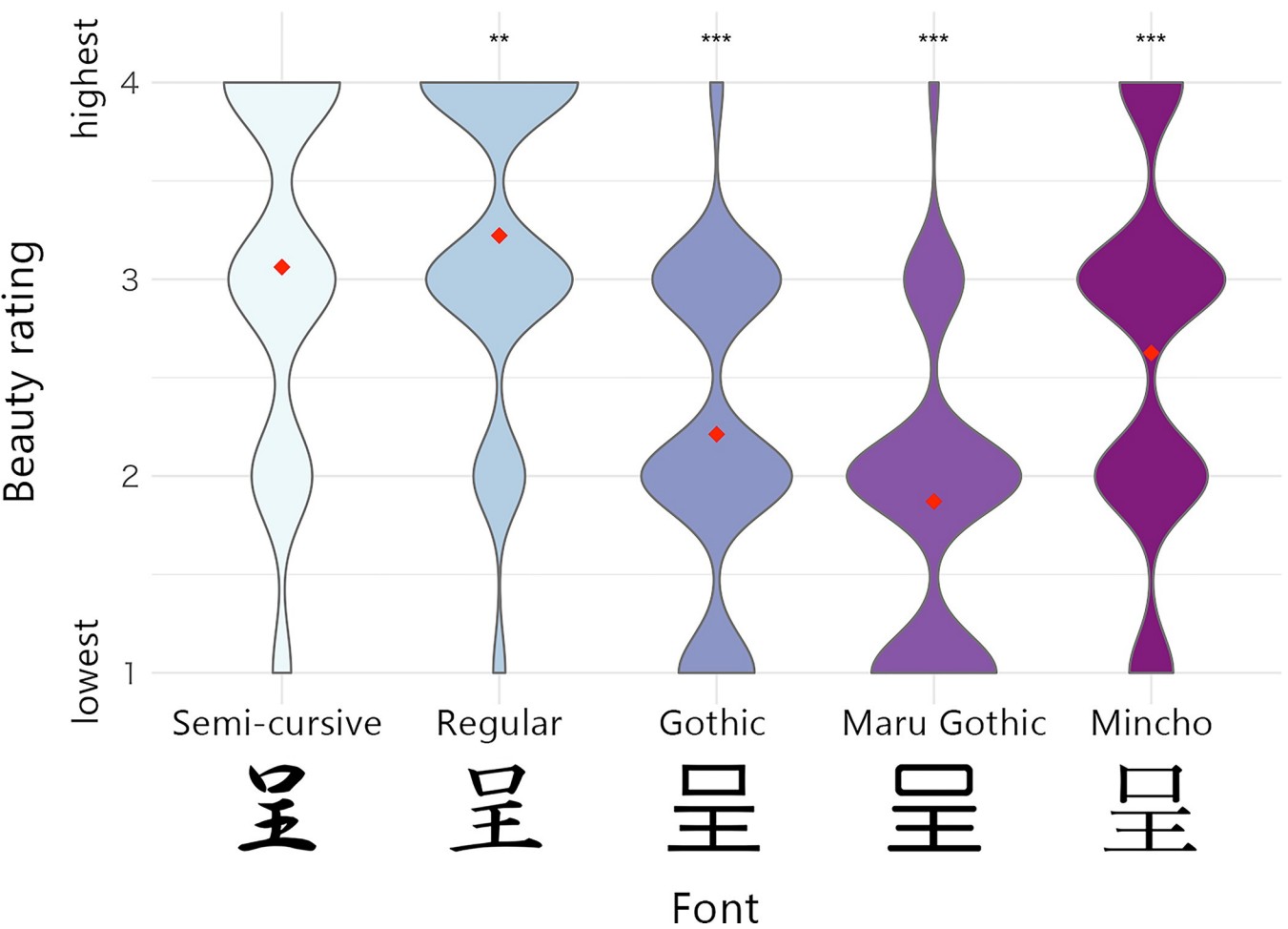

**Fig 2. Stimulus examples and the aesthetic evaluation distributions by typing fonts in Experiment 1.** Note: Beauty ratings are shown separately for each font. The five characters shown at the bottom of this figure are examples of the 90 typing font stimuli used in Experiment 1. Each character represents "present" in Chinese and Japanese. Red points represent mean data. The width of violin plots represents relative frequencies of beauty rating values. The asterisks indicate the GLMM results (Table 1). ***p < .001; **p < .01.

The dependent variable was aesthetic evaluation, and the independent variables were symmetry, complexity, prototypicality, and font type. The five fonts were categorical dummy variables and the Semi-cursive font was treated as the reference. Aesthetic evaluation, symmetry, complexity, and prototypicality were treated as continuous variables. Assuming a Gaussian distribution, we first transformed each continuous variable into a z-score (mean = 0, SD = 1). To control for individual differences among the participants, they were modeled as random effects (intercepts). The perceived properties (symmetry, complexity, and prototypicality) and fonts are fixed effects.

To construct the models, we tested whether treating participants as random effects increased the goodness of fit using the likelihood ratio test (LRT) with α = .05. We compared a fixed intercept model with no independent variables to a mixed model incorporating participants as random effects.

Because the influence of perceived properties on aesthetic evaluation may also differ depending on the font type, we conducted a GLMM for each font separately and added interactions between each structural feature (symmetry, complexity, and prototypicality) and font to the GLMM model. Models with and without interactions were compared using LRT with α = .05.

## Results

Data from one participant were excluded from the analysis because all the responses in the prototypicality task were identical. Data from the remaining 34 participants were analyzed.

**Perceived properties and aesthetic evaluation.** To assess the improvement in goodness of fit, we compared a fixed intercept model with no explanatory variables to a mixed model that included participants as random effects, using LRT. The mixed model showed a better fit ($\chi^2$ = 1384, p < .001).

Table 1 shows the results of the GLMM for the aesthetic evaluation, with the effects of perceived properties and fonts. The first four rows of the table under fixed effects represent the coefficients of the measures belonging to the Semi-cursive script, and the last four rows show the variations in these values for the other fonts with respect to the Semi-cursive script. The GLMM without interaction showed that, while symmetry was not significant ($\beta$ = 0.0168, p = .28), complexity ($\beta$ = 0.0931, p < .001) and prototypicality ($\beta$ = 0.100, p < .001) were significant. These results suggest that aesthetic evaluation did not change as symmetry varied, but was rated higher as complexity and prototypicality increased. Fig 3 shows a summary of the relationship between the perceived properties and aesthetic evaluation.

**Interaction between fonts and perceived properties.** The GLMM revealed that all font effects were significant (ps < .005). As shown in Fig 2, the Regular script was rated higher than the Semi-cursive script, whereas the other fonts were rated lower. We analyzed the aesthetic evaluation of each font separately using the GLMM (S2 File). Only prototypicality had a significant main effect ($\beta$ = 0.1036, p < .001) for the Semi-cursive script. There was no significant main effect for the Regular script. For the Gothic script, there were significant effects of both complexity ($\beta$ = 0.1047, p < .0001) and prototypicality ($\beta$ = 0.08829, p < .01). In the Maru Gothic script, only the effect of complexity was significant ($\beta$ = 0.09809, p < .01). In the Mincho script, both complexity ($\beta$ = 0.1213, p < .001) and prototypicality ($\beta$ = 0.1128, p < .01) were significant. As these results show that the aesthetic evaluations of each font could be influenced by different perceived properties, we conducted a GLMM with interactions between fonts and perceived properties.

**Table 1. Results of the GLMM in Experiment 1 (without interaction).**

| Random effects | | | | |
|---|---|---|---|---|
| | Name | Variance | SD | |
| Participant | (Intercept) | 0.1549 | 0.3936 | |
| Residual | | 0.5741 | 0.7577 | |
| Fixed effects | | | | |
| | Estimate | SE | t value | Pr(>|t|) |
| (Intercept) | 0.4909 | 0.07476 | 6.566 | 4.22e-08*** |
| symmetry | 0.01681 | 0.01551 | 1.083 | 0.2787 |
| complexity | 0.09313 | 0.01650 | 5.643 | 1.83e-08*** |
| prototypicality | 0.1004 | 0.01670 | 6.014 | 2.03e-09*** |
| Regular | 0.1399 | 0.04479 | 3.123 | 0.00181** |
| Gothic | -0.8971 | 0.04523 | -19.84 | <2e-16*** |
| Maru Gothic | -1.207 | 0.04470 | -27.01 | <2e-16*** |
| Mincho | -0.4901 | 0.04557 | -10.76 | <2e-16*** |

Note: **p < .01

***p < .001.

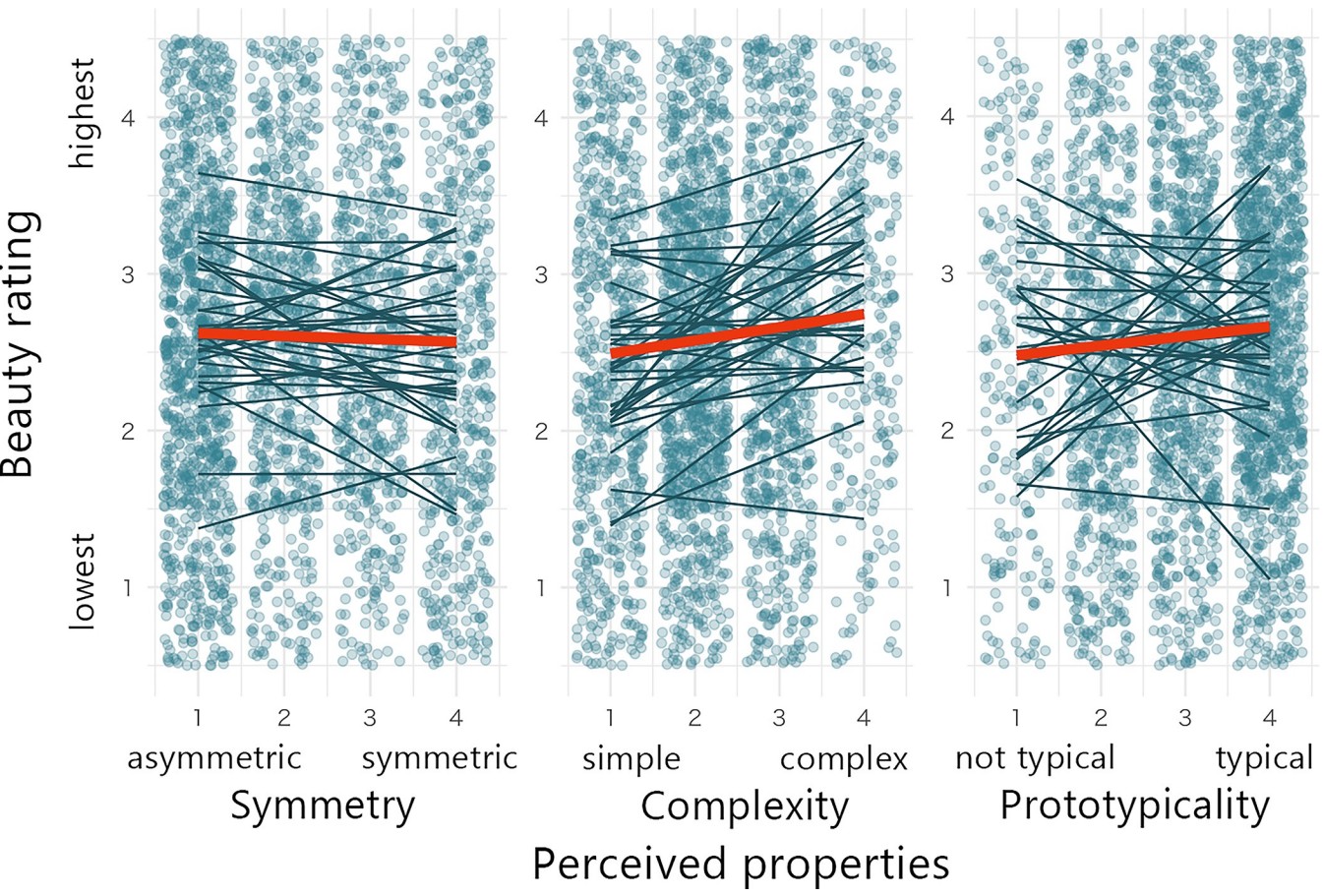

**Fig 3. Results of Experiment 1 for the effect of perceived properties on aesthetic evaluation** (N = 34). Note: Beauty ratings are shown as a function of symmetry, complexity, and prototypicality ratings. Red lines represent the regression results for the mean data. Dark lines represent the regression results for individual participants. For illustrative purposes, overlaid points represent individual settings jittered within areas of ±0.5.

We compared the GLMMs with and without interactions using LRT and found no significant differences between the two models, suggesting that interactions between perceived properties and fonts may not be strong evidence that have influences on aesthetic evaluation ($\chi^2$ = 8.937, p = .708). Table 2 shows that there was no significant effect of complexity for the reference font the Semi-cursive script ($\beta$ = -0.03832, p = .237), while the effect of prototypicality was significant ($\beta$ = 0.1325, p < .001). For the interaction term between complexity and font, the difference between each font and criterion font was significant (Regular script: $\beta$ = 0.09248, p < .05; Gothic script: $\beta$ = 0.1801, p < .001; Maru Gothic script: $\beta$ = 0.2114, p < .001; Mincho script: $\beta$ = 0.1633, p < .001). These results suggest that the effect of complexity depends on the font, and for fonts other than the Semi-cursive script, stimuli with higher complexity are evaluated as more aesthetic. The interaction term for prototypicality was significant only for the Regular script ($\beta$ = -0.1360, p < .01). Considering the slope of prototypicality was close to zero for the Regular script ($\beta$ = -0.0035), prototypicality had little effect on the aesthetic evaluation of the Regular script.

## Discussion

Experiment 1 demonstrated that (a) aesthetic evaluation differed across the fonts (Fig 2) and (b) the influence of perceived properties on aesthetic evaluation also differed across the fonts

**Table 2. Results of the GLMM applied to the aesthetic evaluation in Experiment 1 (with interaction).**

| Random effects | | | | |
|---|---|---|---|---|
| | Name | Variance | SD | |
| Participant | (Intercept) | 0.03587 | 0.1894 | |
| Residual | | 0.1123 | 0.3351 | |
| Fixed effects | | | | |
| | Estimate | SE | t value | Pr(>\|t\|) |
| (Intercept) | 0.9304 | 0.06135 | 15.17 | < 2e-16 *** |
| Symmetry | 0.04243 | 0.02775 | 1.529 | 0.1264 |
| Complexity | 0.007888 | 0.03650 | 0.216 | 0.8289 |
| Prototypicality | 0.1278 | 0.03085 | 4.144 | 3.50e-05 *** |
| Regular | 0.1602 | 0.07630 | 2.099 | 0.03588 * |
| Gothic | -0.4841 | 0.08032 | -6.027 | 1.87e-09 *** |
| Maru Gothic | -0.6359 | 0.07236 | -8.789 | < 2e-16 *** |
| Mincho | -0.3069 | 0.08518 | -3.603 | 0.000320 *** |
| Symmetry:Regular | -0.02546 | 0.03853 | -0.661 | 0.5088 |
| Symmetry:Gothic | -0.03159 | 0.03838 | -0.823 | 0.4105 |
| Symmetry:Maru Gothic | -0.02839 | 0.03794 | -0.748 | 0.4543 |
| Symmetry:Mincho | -0.03132 | 0.03.800 | -0.824 | 0.4100 |
| Complexity:Regular | 0.04988 | 0.04.901 | 1.018 | 0.3089 |
| Complexity:Gothic | 0.1428 | 0.04.913 | 2.907 | 0.003676 ** |
| Complexity:Maru Gothic | 0.1823 | 0.04.920 | 3.706 | 0.000215 *** |
| Complexity:Mincho | 0.1019 | 0.04.876 | 2.089 | 0.03679 * |
| Prototypicality:Regular | -0.1401 | 0.04.6282 | -3.028 | 0.002484 ** |
| Prototypicality:Gothic | 0.01609 | 0.05.129 | 0.314 | 0.7537 |
| Prototypicality:Maru Gothic | -0.01156 | 0.04.642 | -0.249 | 0.8034 |
| Prototypicality:Mincho | 0.03259 | 0.05.464 | 0.597 | 0.5509 |

Note: Colons indicate interactions. ***p < 0.001

**p < 0.01

*p < 0.05.

(Table 2). The aesthetic evaluation of the Semi-cursive and Regular fonts was higher than that of the Gothic, Maru Gothic, and Mincho fonts. This may be because both the Semi-cursive and Regular fonts, which are also used in calligraphy, are more artistic than the other fonts designed to be easily visible and readable in digital formats (Gothic, Maru Gothic, and Mincho) [23]. Complexity increased the aesthetic evaluation of the Gothic, Maru Gothic, and Mincho scripts, whereas prototypicality influenced the Semi-cursive, Gothic, Maru Gothic, and Mincho scripts. Considering that easy-to-read fonts consisting of straight lines are more similar to geometrical patterns than Regular and Semi-cursive scripts, the effect of complexity is consistent with previous studies that have manipulated the complexity of geometrical patterns [4].

## Experiment 2

The purpose of Experiment 2 was to examine whether the font-dependent aesthetic evaluations found in Experiment 1 could be replicated for scripts used in Chinese calligraphy artworks. There are several traditional scripts commonly used in Chinese calligraphy. Whereas the Semi-cursive and Regular scripts are popular in Chinese and Japanese calligraphy artworks [14,15], other scripts with unique styles (such as Cursive, Clerical, and Seal) have also been

appreciated in East Asian cultures [24]. If Chinese character users are sensitive to perceptual differences in calligraphic style, aesthetic evaluation would differ systematically among the distinctive scripts.

## Methods

The methods for Experiment 2 were identical to those of Experiment 1, with the exceptions described below.

**Participants.** We calculated the sample size in the same way as Experiment 1 and the power analysis indicated that at least 32 participants were required (d = 0.4, α = .05, 1−β = .8). Because we planned Experiment 2 as an online experiment instead of a laboratory experiment, we decided to collect more data than the calculated sample size to mitigate the risk of fake data or database error. One hundred Japanese participants (16 female, 49.5 years old, SD = 12.1) were recruited using Yahoo! Crowdsourcing from November 13 to 27, 2022 and paid 100 yen for their participation. Informed consent was obtained from all the participants.

**Apparatus and materials.** The experiment was conducted on the participants' personal computers. A set of 18 the Chinese characters used in Experiment 1 was extracted from calligraphic handwriting written or created around A.D. 25–1322 (http://www.sfds.cn/). Fig 4 shows an example of the five scripts we used, which are representative of Chinese calligraphy and appear in modern copybooks in Japan [24]. Semi-cursive (written by 趙孟頫, Zhao Mengfu), Regular (written by 顔真卿, Yan Zhenqing), Cursive (written by 懷素, Huaisu and 王羲之, Wang Xizhi), Clerical (written on 東漢石碑, Stele in Eastern Han Dynasty and 簡帛, Jianbo), and Seal (collected in 説文解字, Shuowen Jiezi) (Fig 4). Regular is a standard and formal script characterized by its clear and structured appearance. Cursive is a fluid style with simplified flowing strokes and connected elements. Semi-cursive is a script style between Regular and Cursive. Clerical is an ancient style featuring distinct and angular strokes. Seal is the oldest standardized Chinese script, characterized by its square composition with round corners. Each character was approximately 200 pixels in height.

**Procedure.** The participants made two types of judgements (aesthetic and perceived property ratings) in separate blocks in the same manner as in Experiment 1. Before the experimental trials, the participants were asked to respond to a question about their experience training in Chinese or Japanese calligraphy. To detect participants who were unlikely to concentrate on the task, we prepared highly symmetrical and simple character stimuli (大 and 十 presented in the Maru-Gothic font) as catch trials. If participants evaluated these characters as "1 = asymmetric" or "4 = complex," their data were excluded from the analysis.

**Analysis.** To investigate the factors contributing to the aesthetic evaluation of stimuli presented in calligraphic handwriting scripts, we analyzed the data in the same way as in Experiment 1, using the GLMM on aesthetic evaluation as a dependent variable with symmetry, complexity, prototypicality, font type, and calligraphy training as independent variables. In Experiment 2, we used a calligraphic Semi-cursive script as a reference. Participants were modeled as random effects (intercept) and perceived properties (symmetry, complexity, and prototypicality), font, and calligraphy training were modeled as fixed effects.

## Results

Data from three participants could not be retrieved because of network issues, 13 participants failed to pass the catch trials, and four participants had unreasonably short response times of less than 200 ms. Consequently, data from the remaining 80 participants were included in the analysis. Fig 5 shows the overall relationship between the perceived properties and aesthetic evaluations.

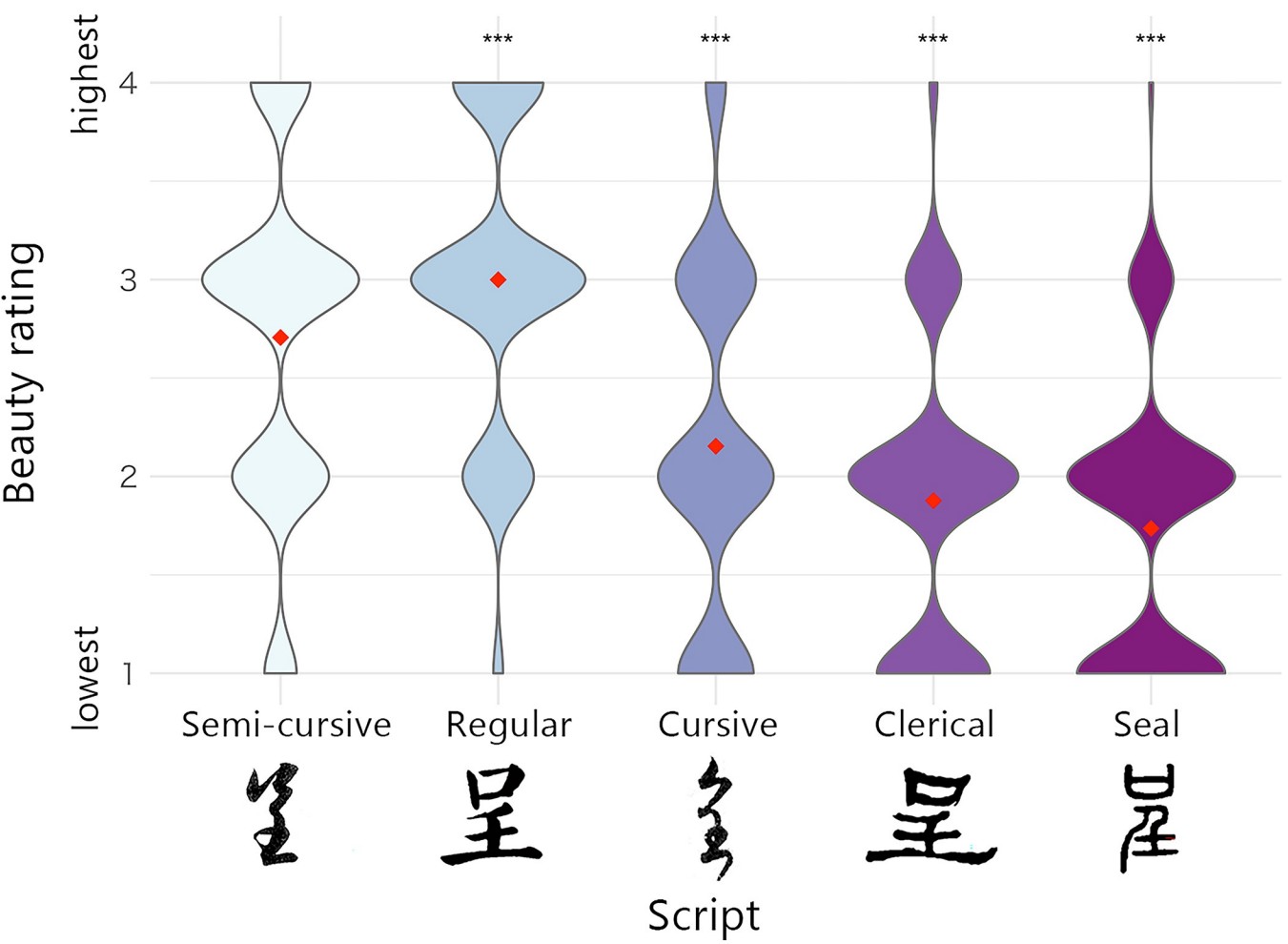

**Fig 4. Stimulus examples and the aesthetic evaluation distributions by calligraphic scripts in Experiment 2.** Note: Beauty ratings are shown separately for each script. The five characters shown at the bottom of this figure are examples of the 90 handwriting script stimuli used in Experiment 2. Each character represents "present" in Chinese and Japanese. Red points represent mean data. The width of violin plots represents relative frequencies of beauty rating values. Asterisks indicate GLMM (Table 3). ***p < .0001.

To assess the potential improvement in goodness of fit, we compared a fixed intercept model with no explanatory variables to a mixed model that included participants as random effects, using LRT. The mixed model showed a better fit ($\chi^2$ = 1111, p < .001). Table 3 shows a summary of the GLMM (without interaction) for aesthetic evaluation. The first four rows of the table under fixed effects represent the coefficients of the measures obtained for the Semi-cursive script without calligraphic training. The remaining five rows show the variations in these values for the other variables with respect to the Semi-cursive script.

The GLMM without interaction revealed that prototypicality ($\beta$ = 0.1800, p < .001) showed a significant effect on aesthetic evaluation, while neither symmetry ($\beta$ = 0.01354, p = .169), complexity ($\beta$ = 0.001427, p = .894), nor calligraphic training ($\beta$ = 0.06742, p = .151) demonstrated significance. That is, neither symmetry nor complexity influenced aesthetic evaluation, whereas prototypicality had a positive effect on aesthetic evaluation. All the effects of the fonts (intercepts) were also significant (ps < .005).

**Interaction between fonts and perceived properties.** Fig 4 summarizes the mean aesthetic evaluations for each script. As revealed by the results of the GLMM (Table 3), the

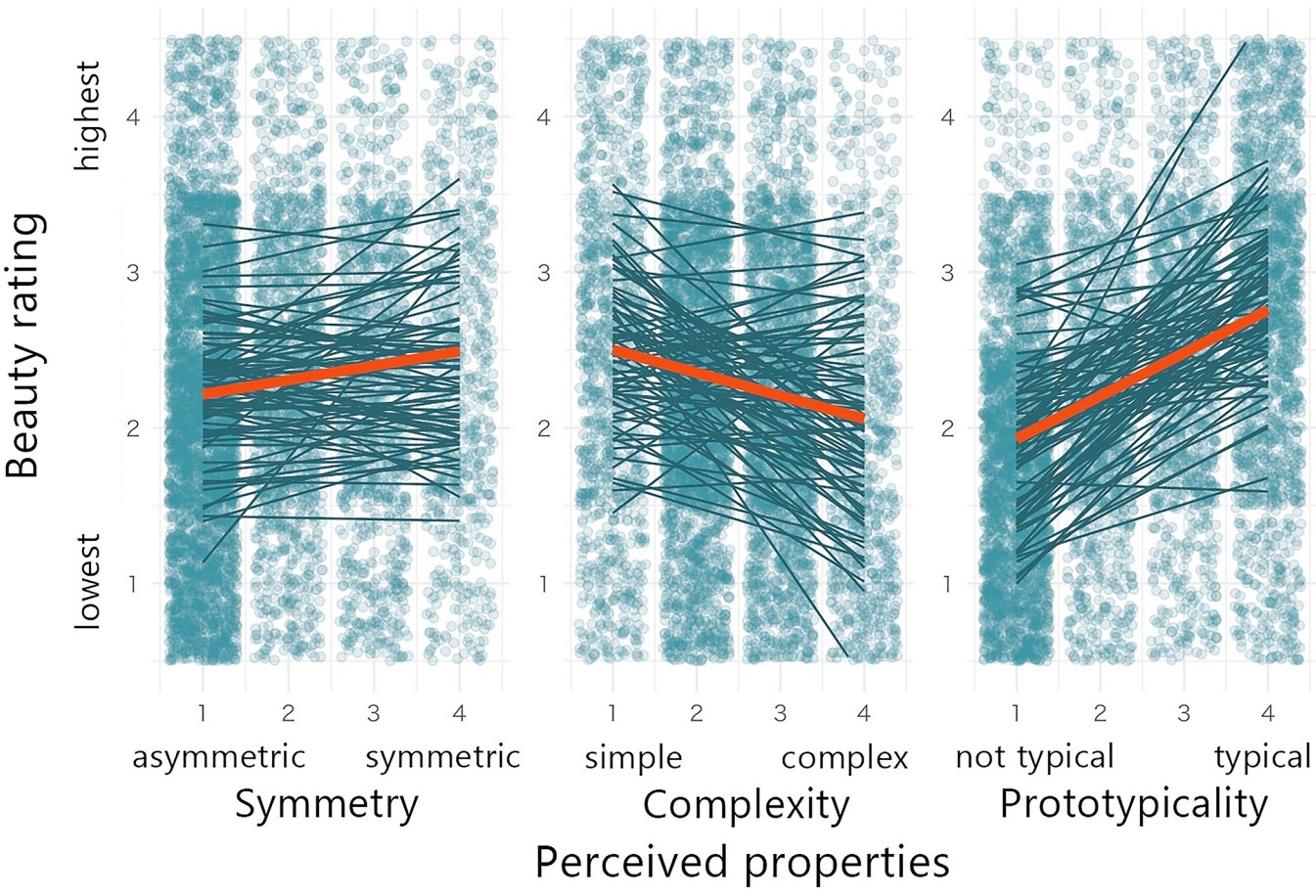

**Fig 5. Results of Experiment 2 for the effect of perceived properties on aesthetic evaluation (N = 80).** Note: Beauty ratings are shown as a function of symmetry, complexity, and prototypicality ratings. Red lines represent the regression results for the mean data. Dark lines represent the regression results for individual participants. For illustrative purposes, overlaid points represent individual settings jittered within areas of ±0.5.

aesthetic evaluation of the Regular script was significantly higher than that of the Semi-cursive script, whereas the aesthetic evaluation of the Cursive, Clerical, and Seal scripts was lower than that of the Semi-cursive script.

We analyzed the aesthetic evaluations using the GLMM for each script in the same manner as in Experiment 1. For the Semi-cursive script, only prototypicality was significant (β = 0.2419, p < .001). For the Regular script, only prototypicality was significant (β = 0.1300, p < .001). For the Cursive script, both complexity (β = -0.8247, p < .001) and prototypicality (β = 0.2519, p < .001) were significant. For the Clerical script, symmetry (β = 0.08825, p < .001) and prototypicality (β = 0.08100, p < .01) were significant. For the Seal script, both complexity (β = 0.03628, p < .05) and prototypicality (β = 0.09133, p < .001) were significant. The results for each font are shown in S3 File.

As in Experiment 1, we conducted a GLMM with interactions between the scripts and perceived properties. We compared the GLMMs with and without interactions using LRT. In contrast to Experiment 1, the GLMM with interaction showed a better fit ($\chi^2$ = 30.58, p < .05). Table 4 summarizes the GLMM with the interactions. For the reference Semi-cursive script, only prototypicality was significant (β = 0.2489, p < .001). The interaction between symmetry and script was significant for the Clerical script (β = 0.0850, p < .01), indicating that the aesthetic evaluation of this script was influenced by symmetry. The interaction between

**Table 3. Results of the GLMM applied to the aesthetic evaluation in Experiment 2 (without interaction).**

| Random effects | | | | |
|---|---|---|---|---|
| | Name | Variance | SD | |
| Participant | (Intercept) | 0.1666 | 0.4082 | |
| Residual | | 0.5373 | 0.7330 | |
| Fixed effects | | | | |
| | Estimate | SE | t value | Pr(>\|t\|) |
| (Intercept) | 0.4186 | 0.04958 | 8.443 | 2.34e-13 *** |
| Symmetry | 0.01354 | 0.009.833 | 1.377 | 0.169 |
| Complexity | 0.001427 | 0.01074 | 0.133 | 0.894 |
| Prototypicality | 0.1810 | 0.01241 | 14.59 | < 2e-16 *** |
| Regular | 0.1726 | 0.02885 | 5.983 | 2.29e-09 *** |
| Cursive | -0.4719 | 0.02841 | -16.61 | < 2e-16 *** |
| Clerical | -0.8982 | 0.02741 | -32.77 | < 2e-16 *** |
| Seal | -0.8957 | 0.02909 | -30.79 | < 2e-16 *** |
| Training | 0.06742 | 0.04647 | 1.451 | 0.151 |

Note: ***p < 0.001.

complexity and script was also significant for the Clerical script (β = 0.0624, p < .05). The interaction between prototypicality and script was significant for the Regular (β = -0.1334, p < .001), Clerical (β = -0.1566, p < .001), and Seal (β = -0.1357, p < .001) scripts relative to the reference script (Semi-cursive). Considering the signs of the regression coefficients (i.e., slopes), it can be interpreted that the influence of prototypicality on the Regular, Clerical, and Seal scripts (all negative slopes) was smaller than that on the Semi-cursive script (positive slope).

## Discussion

Similar to typing fonts used in Experiment 1, the aesthetic evaluation of Chinese characters differed across calligraphic scripts with interactions with perceived properties. As in Experiment 1, the Regular and Semi-cursive scripts were preferred over the other scripts. These results are generally consistent with those of Xu and Shen, who demonstrated that neither Chinese nor non-Chinese participants preferred the Seal or Cursive scripts [25]. Furthermore, Xu and Shen demonstrated that non-Chinese character users preferred running scripts (somewhat similar to the Semi-cursive script) to the Regular script. Therefore, we speculate that, possibly through experience, higher aesthetic evaluations are produced by Chinese character users who perceive the Regular and Semi-cursive scripts as containing more artistic features than other scripts.

Supplementary analyses (S3 File) revealed that the aesthetic evaluation of the Seal script was positively correlated with complexity, whereas the complexity effect was negative for the Cursive script. The direction of the effect of complexity was different. The symmetry effect was significant only for the Clerical script, with a positive slope. The exact reason for this result is not entirely clear; strokes in the Clerical script are clearly bolder than those in other scripts (Fig 4), yielding a different result.

## General discussion

This study investigated the relationship between perceived properties and the aesthetic evaluation of meaningful Chinese characters shown in typing fonts (Experiment 1) and calligraphic

**Table 4. Results of the GLMM applied to the aesthetic evaluation in Experiment 2 (with interaction).**

| Random effects | | | | |
|---|---|---|---|---|
| Groups | Name | Variance | SD | |
| Participant | (Intercept) | 0.1455 | 0.3815 | |
| Residual | | 0.4593 | 0.6777 | |
| Fixed effects | | | | |
| | Estimate | SE | t value | Pr(>\|t\|) |
| (Intercept) | 2.103 | 0.1298 | 16.20 | < 2e-16 *** |
| Symmetry | 0.01667 | 0.01879 | 0.887 | 0.3752 |
| Complexity | -0.0006342 | 0.02302 | -0.028 | 0.9780 |
| Prototypicality | 0.1909 | 0.01693 | 11.27 | < 2e-16 *** |
| Regular | 0.6463 | 0.1380 | 4.685 | 2.85e-06 *** |
| Cursive | -0.3237 | 0.1279 | -2.531 | 0.011405 * |
| Clerical | -0.8758 | 0.1280 | -6.840 | 8.57e-12 *** |
| Seal | -0.6843 | 0.1293 | -5.292 | 1.24e-07 *** |
| Training | 0.04265 | 0.03634 | 1.174 | 0.2433 |
| Symmetry:Regular | -0.04095 | 0.02513 | -1.630 | 0.1032 |
| Symmetry:Cursive | -0.01109 | 0.02783 | -0.398 | 0.6904 |
| Symmetry:Clerical | 0.06763 | 0.02526 | 2.678 | 0.007434 ** |
| Symmetry:Seal | -0.03113 | 0.02492 | -1.249 | 0.2116 |
| Complexity:Regular | -0.01705 | 0.03195 | -0.534 | 0.5935 |
| Complexity:Cursive | -0.04647 | 0.03130 | -1.485 | 0.1376 |
| Complexity:Clerical | 0.06213 | 0.03171 | 1.960 | 0.05008 |
| Complexity:Seal | 0.02582 | 0.03084 | 0.837 | 0.4027 |
| Prototypicality:Regular | -0.1064 | 0.02776 | -3.832 | 0.000128 *** |
| Prototypicality:Cursive | 0.03133 | 0.02636 | 1.188 | 0.2347 |
| Prototypicality:Clerical | -0.1169 | 0.02402 | -4.868 | 1.15e-06 *** |
| Prototypicality:Seal | -0.1036 | 0.02728 | -3.796 | 0.000148 *** |
| Regular:Training | -0.02444 | 0.02009 | -1.216 | 0.2239 |
| Cursive:Training | 0.004671 | 0.02004 | 0.233 | 0.8157 |
| Clerical:Training | 0.02804 | 0.02003 | 1.400 | 0.1615 |
| Seal:Training | 0.01859 | 0.01997 | 0.931 | 0.3518 |

Note: Colons indicate interactions. ***p < 0.001

**p < 0.01

*p < 0.05.

handwriting scripts (Experiment 2). The main findings are summarized as follows: (a) aesthetic evaluation was correlated with prototypicality for many fonts and scripts; (b) aesthetic evaluation was partly correlated with complexity only for typing fonts, and no correlation was found between aesthetic evaluation and symmetry in typing fonts or handwriting scripts. Therefore, the effects of perceived properties on the aesthetic evaluation of Chinese characters are partially consistent with those of geometrical figures reported in previous studies.

In both Experiments 1 and 2, the participants generally preferred the Semi-cursive and Regular scripts. Since the two scripts contain moderately curved strokes (Figs 2 and 4), these results seem to be explained partly by a general preference for objects with round corners over those with sharp edges [26]. However, curvature itself is insufficient for explaining all results since some of the other scripts have clearly round edges (e.g., the Maru Gothic, Cursive, Clerical, and Seal scripts).

## Influence of perceived properties on aesthetic evaluation of Chinese characters

**Prototypicality.** We found a positive correlation between aesthetic evaluation and proto-typicality across the two experiments, especially for the Semi-cursive scripts. This is generally consistent with the results obtained from geometrical figures [5] and Chinese characters shown to non-Chinese character users [15]. As Whitfield and Slatter noted, prototypicality is partly formed by familiarization [5]. Park et al. and Liao et al. reported that familiarization increases the rate of preference for faces but decreases the rate of preference for natural scenes [9,10]. Thus, the mechanisms underlying the aesthetic evaluation of characters may be similar to those of faces rather than natural scenes.

A reviewer asked as to consider the possibility that the association between prototypicality and aesthetic evaluation may be explained by lexical access to meaning represented by the characters. As an extreme case, consider that the meaning of all stimuli is positive. In this case, prototypical fonts or scripts would produce a higher aesthetic evaluation as a consequence of the metacognitive decision based on the stimulus content [2]. However, this possibility is unlikely since we selected and used character stimuli whose mean emotional valence was very close to neutral (mean = 2.06 on a rating scale of 1 to 3; Fig 1). Therefore, the present data provide evidence for Reber et al.'s idea that prototypicality increases aesthetic evaluation independently of the stimulus content [2].

**Complexity.** There was a significant linear association between perceived complexity and the aesthetic evaluation of Chinese characters in typing fonts (Experiment 1), but not in calligraphic scripts (Experiment 2). The results of Experiment 1 were generally consistent with previous results obtained using geometrical patterns [1]. Similar to the results of Experiment 2, Han et al. found that calligraphy artworks (particularly those drawn in running scripts) were evaluated as aesthetic only when complexity was moderate [27]. Such a nonlinear relationship between complexity and aesthetic evaluation might underlie the earlier finding that complexity is not associated with the aesthetic evaluation of Japanese calligraphy by non-Chinese-character users [15]. As Fillinger and Hubner used character stimuli extracted from calligraphic artworks, their stimuli resembled calligraphic scripts (Experiment 2 and Han et al. [27]) rather than typing fonts (Experiment 1) [15]. Therefore, the discrepant results between Experiments 1 and 2 may correspond to previous findings on geometrical patterns [1] and calligraphic artworks [15], respectively. Taken together, our results demonstrate that script style plays an important role in understanding the association between the complexity and aesthetic evaluation of Chinese characters, even in Chinese character users who extract meaning from the stimuli.

**Symmetry.** There was no significant linear association between perceived symmetry and the aesthetic evaluation of Chinese characters represented by either typing (Experiment 1) or calligraphic scripts (Experiment 2). These results were unexpected, given the previous finding that aesthetic evaluation of symmetrical geometrical patterns was higher than that of asymmetrical patterns [1]. To reconcile the present data with previous findings, we considered the following three possibilities. First, the variation in the perceived symmetry in our experiments may have been too small to detect a significant effect. We argue that this is unlikely because the s.d. of symmetry ratings (1.13 and 1.05 in Experiments 1 and 2, respectively) was comparable to that of the other perceived properties, that is, complexity (0.93 and 0.91 in Experiments 1 and 2, respectively) and prototypicality (0.99 and 1.21 in Experiments 1 and 2, respectively). Second, the participants' interpretation of "symmetry" in our stimulus set of Chinese characters might differ from that of geometrical patterns, as in the case of interpretations of "balance" that depend on stimulus category [11]. Notably, symmetry ratings were not required in

previous studies [1,11,28]. Third, Gartus and Leder reported that nearly symmetrical patterns produced lower preference ratings, probably reflecting a nonlinear relationship between symmetry and aesthetic evaluations [28]. Indeed, some of the Chinese characters used here were nearly symmetrical (e.g., perfectly symmetrical, except for the right ends of the Mincho script in Fig 2). Therefore, we might not find a reliable linear effect of perceived symmetry on aesthetic evaluation. Taken together, to comprehensively understand the psychological mechanisms underlying the aesthetic evaluation of Chinese characters, it would be useful to (a) increase and balance stimulus lineups in terms of symmetry, (b) include balance ratings in addition to symmetry ratings [15], and (c) consider objective measures for symmetry and balance (e.g., center of mass [29]) in future studies.

## Implications for aesthetic evaluation of other materials

Recent studies using several types of visual pattern have indicated unique roles of familiarization on aesthetic evaluation [1,9]. As described in the introduction, familiarization is involved in the formation of prototypicality. Positive correlations between prototypicality and aesthetic evaluation found for several scripts (e.g., the Semi-cursive scripts) are generally in line with the previous studies using face images [9,10]. However, no significant negative correlation was found between prototypicality and aesthetic evaluation for any script. Notably, negative correlations between familiarization and aesthetic evaluation have been reported for landscape images [9,10] and geometrical patterns [1], indicating a positive contribution of novelty on aesthetic evaluation. Considering the previous and present results, we speculate that faces and meaningful Chinese characters are aesthetically evaluated by familiarity-based mechanisms, whereas landscape and geometrical figures are evaluated by novelty-based mechanisms.

## Conclusions

This study systematically investigated the role of perceived symmetry, complexity, and prototypicality in the aesthetic evaluation of Chinese characters using several typing fonts and calligraphic scripts. By analyzing the relative impacts of the perceived properties on aesthetic evaluation and their interactions with script style, we discovered that script style plays an important role in the association between the perceived properties and aesthetic evaluation of Chinese characters. Some of our results are consistent with insights obtained from the aesthetic evaluation of geometrical figures. More detailed analyses will help reveal the aesthetic and practical aspects of Chinese characters.

## Supporting information

**S1 File. Pilot experiment.** Procedure of pilot experiment.
(DOCX)

**S2 File. Results of the GLMM applied to the aesthetic evaluation in Experiment 1 for each font.**
(DOCX)

**S3 File. Results of the GLMM applied to the aesthetic evaluation in Experiment 2 for each script.**
(DOCX)

## Acknowledgments

We thank Daiichiro Kuroki for assistance in computer programming and Editage (www.editage.com) for English language editing.

## Author Contributions

**Conceptualization:** Qinjing Zhang.

**Formal analysis:** Qinjing Zhang, Hiroyuki Mitsudo.

**Funding acquisition:** Qinjing Zhang, Hiroyuki Mitsudo.

**Investigation:** Qinjing Zhang.

**Methodology:** Qinjing Zhang, Hiroyuki Mitsudo.

**Project administration:** Hiroyuki Mitsudo.

**Supervision:** Hiroyuki Mitsudo.

**Visualization:** Qinjing Zhang, Hiroyuki Mitsudo.

**Writing – original draft:** Qinjing Zhang, Hiroyuki Mitsudo.

**Writing – review & editing:** Qinjing Zhang, Hiroyuki Mitsudo.

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
