## [Decision Letter · Decision Letter 0]

11 Jul 2024

PONE-D-24-04493Aesthetic evaluation and the perceived properties of Chinese charactersPLOS ONE

Dear Dr. Zhang,

Thank you for submitting your manuscript to PLOS ONE. After careful consideration, we feel that it has merit but does not fully meet PLOS ONE’s publication criteria as it currently stands. Therefore, we invite you to submit a revised version of the manuscript that addresses the all the points raised during the review process.

We look forward to receiving your revised manuscript.

Kind regards,

J Malo

Academic Editor

PLOS ONE

Journal Requirements:

Reviewers' comments:

Reviewer's Responses to Questions

**Comments to the Author**

1. Is the manuscript technically sound, and do the data support the conclusions?

Reviewer #1: Yes

2. Has the statistical analysis been performed appropriately and rigorously? 

Reviewer #1: Yes

3. Have the authors made all data underlying the findings in their manuscript fully available?

Reviewer #1: Yes

4. Is the manuscript presented in an intelligible fashion and written in standard English?

Reviewer #1: Yes

5. Review Comments to the Author

Reviewer #1: Review of Aesthetic Evaluation and the Perceived Properties of Chinese Characters

Qinjing Zhang and Hiroyuki Mitsudo

The psychological research on complexity is much more nuanced. It is an oversimplification to say that 'aesthetic evaluation improves when a stimulus is perceived as more complex'. Sometimes there is an inverted-U function, sometimes there are simplicity effects. The literature on this subject needs to be described more fully.

Geometric figures however are also less naturalistic and generalizable than more familiar images like faces and scenes so there is a trade-off between using them and more abstract patterns. The 'Figures and Artworks' sections is under-described. A few studies are cited, but towards what end, to justify use of characters? Familiarization effects and number of elements are described but the purpose of this section needs to me made more explicit.

Do ideographs also look like the things they represesent? Any predicted differences in interpretation or preference for symbolic, phonetic, pictographic or hybrid forms?

Predictions regarding meaning in characters? How might meaning affect symmetry, complexity, and prototypicality compared to purely geometric patterns without semantics?

Why only five characters? They all represent grasses and so are controlled for meaning but doesn't this severely limit the generalizability of the results? Given the large number of such characters used in an alphabet, isn't this quite limiting?

Motivations for using the different fonts are needed. How did these differ from one another structurally and stylistically and why were they chosen?

How were the various metrics calculated? There are three types of symmetry and within these various quantitative amounts. I don't see a detailed mention of this.

Cursive fonts contain curvature, which is preferred more than straight lines. There is fairly extensive recent work on these differences, but this is not mentioned.

There is no discussion section after Experiment 1.

What was the motivation for Experiment 2? This should flow consequentially from the meaning of the results of the prior study. It isn't enough to just say these are the methodological differences as described below.

I wonder if prototypicality effects can be explained in terms of facilitating lexical access to stored meaning. A prototype may more easily activated stored meaning than a more average or outlying representation.

Figure 1, y-axis and quantitative aspect of the x-axis not described. Greater bulges are greater variance...

Fully label axes in Figure 2, same going forward for other figures.

How does this work connect to the aesthetic research on geometric figures, artwork, and naturalistic images like faces and landscapes?

6. PLOS authors have the option to publish the peer review history of their article (what does this mean?). If published, this will include your full peer review and any attached files.

Reviewer #1: **Yes: **Jay Friedenberg

---

## [Author Response · Author response to Decision Letter 0]

22 Aug 2024

Response to Reviewer

Thank you very much for providing helpful comments and suggestions on our original manuscript. Based on all the comments made by Reviewer #1, we revised the manuscript. In short, we expanded on the introduction, method, and general discussion sections. Changed parts in the revised manuscript are highlighted by blue text. The following are the comments made by Reviewer #1 and our responses. Page and line numbers indicate those of the revised manuscript.

Comment #1: The psychological research on complexity is much more nuanced. It is an oversimplification to say that 'aesthetic evaluation improves when a stimulus is perceived as more complex'. Sometimes there is an inverted-U function, sometimes there are simplicity effects. The literature on this subject needs to be described more fully.

Response: In response to this comment, we revised the relevant sentence in the first paragraph of the Introduction in accordance with your suggestion (pp. 2-3, lines 37-40). The revised sentences reads: " In contrast to the case of symmetry, the relation between complexity and aesthetic evaluation is less straightforward [2]. Some studies suggest an inverted U-shape relationship between complexity and aesthetic evaluation, where aesthetic evaluation increases when the stimulus complexity is intermediate [3] ."

Comment #2: Geometric figures however are also less naturalistic and generalizable than more familiar images like faces and scenes so there is a trade-off between using them and more abstract patterns. The 'Figures and Artworks' sections is under-described. A few studies are cited, but towards what end, to justify use of characters? Familiarization effects and number of elements are described but the purpose of this section needs to me made more explicit.

Response: In an attempt to clearly explain the background for our stimulus choice, we made two changes in the manuscript. First, we changed the label of the subsection heading (p. 3, line 48) as "Visual object category." Second, we rewrote the relevant section to expand on why we needed a stimulus category where perceived properties and meaning are easily manipulated (p. 3, lines 49-63).

Comment #3: Do ideographs also look like the things they represesent? Any predicted differences in interpretation or preference for symbolic, phonetic, pictographic or hybrid forms?

Response: The ideographs in ancient China were similar to the things represented. However, ideographs we use today is no longer resemble things they represented. As a matter of fact, it was difficult to present specific predictions from the idiographic aspects of Chinese characters. We corrected misleading expressions (p. 4, lines 67-60) and the revised sentences are as follows: ".... Elementary parts of Chinese characters have ideographic origins. Whereas phonetic scripts such as the alphabet only contain sound features, ideographic characters are useful to convey meanings through visual image features. Therefore, Chinese characters can be regarded as stimuli that can not only be easily manipulated as patterns, but also carry meanings similar to visual artworks."

Comment #4: Predictions regarding meaning in characters? How might meaning affect symmetry, complexity, and prototypicality compared to purely geometric patterns without semantics?

Response: Because the explanation about the meaning of our Chinese character stimuli was insufficient, we made three changes in the revised manuscript. In short, we believe that the meaning of Chinese characters itself is unlikely to alter ratings in our experiments. First, we added a sentence stating that we chose characters with relatively neutral emotional valences (p. 5, line 97-99). Second, we added an explanation that no characters explicitly meant symmetry, complexity, prototypicality, or beauty on p. 6 (line 130) as well as a figure showing all character meaning with English translation (Fig 1 and p. 7, lines 138-140). Third, we added a paragraph to discuss the possibility that meaning might affect aesthetic evaluation through lexical access suggested by Comment #11 (p. 21, lines 389-396). 

Comment #5: Why only five characters? They all represent grasses and so are controlled for meaning but doesn't this severely limit the generalizability of the results? Given the large number of such characters used in an alphabet, isn't this quite limiting?

Response: Because our description on the stimuli was misleading, we made several changes in the revised manuscript as follows. The number of characters (with different meanings) were 18 and each of them was presented by five different fonts (in Experiment 1) and calligraphic scripts (in Experiment 2). To show this point explicitly, we added a sentence in the introduction (p. 5, line 99-101) and the new Fig 1 to make it clear enough to understand. Because the five example characters (representing grasses) were inaccurately displayed, they were replaced by other characters (representing “present”; Figs 2 and 4). We also expanded on the note of the new Fig 2 in an attempt to describe the number of characters explicitly (p. 7, lines 143-146).

Comment #6: Motivations for using the different fonts are needed. How did these differ from one another structurally and stylistically and why were they chosen?

Response: We used several fonts to make this study comprehensive since different fonts have been used for various situations in China and Japan. To explain this point, we added sentences on p. 5 (lines 97-99) and p. 7 (lines132-137). Detailed descriptions of the fonts were also provided on p. 7. Similar changes were made for Experiment 2 (p. 14, lines 265-270).

Comment #7: How were the various metrics calculated? There are three types of symmetry and within these various quantitative amounts. I don't see a detailed mention of this.

Response: We did not calculate metrics of image features (i.e., symmetry and complexity). Instead, we asked participants to rate the perceived symmetry, complexity, and prototypicality of each character in Experiments 1 and 2. To make this point more explicit, we modified the relevant descriptions throughout the manuscript including figure labels (e.g., p. 5, line 103 and Figs 3 and 5).

Comment #8: Cursive fonts contain curvature, which is preferred more than straight lines. There is fairly extensive recent work on these differences, but this is not mentioned.

Response: Following this comment, we cited Bar et al. (2006) and added a paragraph to discuss this issue on p. 20-21 (lines 373-377).

Comment #9: There is no discussion section after Experiment 1.

Response: We added a discussion paragraph of Experiment 1 on pp. 13 (lines 227-238) by modifying and moving the paragraph which discussed the results of Experiment 1 from in the General discussion section of the original manuscript.

Comment #10: What was the motivation for Experiment 2? This should flow consequentially from the meaning of the results of the prior study. It isn't enough to just say these are the methodological differences as described below.

Response: Following this comment, we added the preamble of Experiment 2. In short, the purpose of this experiment was to examine whether the font-dependent aesthetic evaluations found in Experiment 1 was replicated for Chinese calligraphy scripts (p. 13, lines 240-246).

Comment #11: I wonder if prototypicality effects can be explained in terms of facilitating lexical access to stored meaning. A prototype may more easily activated stored meaning than a more average or outlying representation.

Response: To discuss this issue, we added a paragraph in the General discussion section (p. 21, lines 389-396). The paragraph reads: "A reviewer asked as to consider the possibility that the association between prototypicality and aesthetic evaluation may be explained by lexical access to meaning represented by the characters. As an extreme case, consider that the meaning of all stimuli is positive. In this case, prototypical fonts or scripts would produce a higher aesthetic evaluation as a consequence of the metacognitive decision based on the stimulus content [2]. However, this possibility is unlikely since we selected and used character stimuli whose mean emotional valence was very close to neutral (mean = 2.06 on a rating scale of 1 to 3; Fig 1). Therefore, the present data provide evidence for Reber et al.’s idea that prototypicality increases aesthetic evaluation independently of the stimulus content [2]."

Comment #12: Figure 1, y-axis and quantitative aspect of the x-axis not described. Greater bulges are greater variance...

Response: We modified the labels of new Fig 2 and added an explanation. The width of violin plots represents relative frequencies of beauty rating values (p. 7, lines 145-146).

Comment #13: Fully label axes in Figure 2, same going forward for other figures.

Response: Following this comment, we added details of axis labels of new Figs 2-5.

Comment #14: How does this work connect to the aesthetic research on geometric figures, artwork, and naturalistic images like faces and landscapes?

Response: To discuss an implication of this study, we added a paragraph in the General discussion section (pp. 23, lines 434-445). The paragraph reads: "Recent studies using several types of visual pattern have indicated unique roles of familiarization on aesthetic evaluation [1,9]. As described in the introduction, familiarization is involved in the formation of prototypicality. Positive correlations between prototypicality and aesthetic evaluation found for several scripts (e.g., the Semi-cursive scripts) are generally in line with the previous studies using face images [9,10]. However, no significant negative correlation was found between prototypicality and aesthetic evaluation for any script. Notably, negative correlations between familiarization and aesthetic evaluation have been reported for landscape images [9,10] and geometrical patterns [1], indicating a positive contribution of novelty on aesthetic evaluation. Considering the previous and present results, we speculate that faces and meaningful Chinese characters are aesthetically evaluated by familiarity-based mechanisms, whereas landscape and geometrical figures are evaluated by novelty-based mechanisms."

---

## [Decision Letter · Decision Letter 1]

15 Jan 2025

Aesthetic evaluation and the perceived properties of Chinese characters

PONE-D-24-04493R1

Dear Dr. Zhang,

We’re pleased to inform you that your manuscript has been judged scientifically suitable for publication and will be formally accepted for publication once it meets all outstanding technical requirements.

Kind regards,

Sadiq H. Abdulhussain, Ph.D.

Academic Editor

PLOS ONE

Additional Editor Comments (optional):

Reviewers' comments:

Reviewer's Responses to Questions

**Comments to the Author**

1. If the authors have adequately addressed your comments raised in a previous round of review and you feel that this manuscript is now acceptable for publication, you may indicate that here to bypass the “Comments to the Author” section, enter your conflict of interest statement in the “Confidential to Editor” section, and submit your "Accept" recommendation.

Reviewer #1: All comments have been addressed

Reviewer #2: All comments have been addressed

Reviewer #3: All comments have been addressed

2. Is the manuscript technically sound, and do the data support the conclusions?

Reviewer #1: Yes

Reviewer #2: Yes

Reviewer #3: Yes

3. Has the statistical analysis been performed appropriately and rigorously? 

Reviewer #1: Yes

Reviewer #2: I Don't Know

Reviewer #3: Yes

4. Have the authors made all data underlying the findings in their manuscript fully available?

Reviewer #1: Yes

Reviewer #2: Yes

Reviewer #3: Yes

5. Is the manuscript presented in an intelligible fashion and written in standard English?

Reviewer #1: Yes

Reviewer #2: Yes

Reviewer #3: Yes

6. Review Comments to the Author

Reviewer #1: (No Response)

Reviewer #2: In this study, an aesthetic evaluation for Chinese characters is presented.

The authors have addressed the raised comments.

Reviewer #3: (No Response)

7. PLOS authors have the option to publish the peer review history of their article (what does this mean?). If published, this will include your full peer review and any attached files.

Reviewer #1: **Yes: **Jay Friedenberg

Reviewer #2: No

Reviewer #3: No

---

## [Editor Report · Acceptance letter]

23 Jan 2025

PONE-D-24-04493R1 

PLOS ONE

Dear Dr. Zhang, 

I'm pleased to inform you that your manuscript has been deemed suitable for publication in PLOS ONE. Congratulations! Your manuscript is now being handed over to our production team.

Kind regards, 

on behalf of

Dr. Sadiq H. Abdulhussain 

Academic Editor

PLOS ONE